DESY-23-195

# Combining QED and Approximate N³LO QCD Corrections in a Global PDF Fit: MSHT20qed_an3lo PDFs

T. Cridge[a], L. A. Harland-Lang[b], and R.S. Thorne[b]

[a] Deutsches Elektronen-Synchrotron DESY, Notkestr. 85, Hamburg 22607, Germany
[b] Department of Physics and Astronomy, University College London, London, WC1E 6BT, UK

## Abstract

We present the MSHT20qed_an3lo parton distribution functions (PDFs). These result from the first global PDF analysis to combine QED and approximate N³LO (aN³LO) QCD corrections in the theoretical calculation of the PDF evolution and cross sections entering the fit. We examine the PDF impact, and find that the effect of QED is relatively mild in comparison to the aN³LO corrections, although it should still be accounted for at the level of precision now required. These QED corrections are in addition found to roughly factorise from the QCD corrections; that is, their relative impact on the PDFs is roughly the same at NNLO and aN³LO. The fit quality exhibits a very small deterioration at aN³LO upon the inclusion of QED corrections, which is rather smaller than the deterioration observed at NNLO in QCD. The impact on several cross sections at N³LO is also examined, including the Higgs cross section via gluon fusion at N³LO. Finally, a LO in QCD fit that includes QED corrections is also presented: the MSHT20qed_lo set.

## 1 Introduction

The high precision requirements of the Large Hadron Collider (LHC) physics programme necessitate a correspondingly high level of precision and accuracy in the determination of the parton distribution functions (PDFs). To achieve this, dedicated global PDF fits are performed by multiple groups [1–3], see [4] for a recent summary. A key element in this is to work with as high precision as possible in the perturbative expansion of the theoretical ingredients entering the fit, from the evolution of the PDFs to the relevant cross section calculations.

Until recently, these PDF fits have been provided to at most next–to–next–to leading order (NNLO) in the QCD perturbative expansion. However, in [5] the first PDF analysis at approximate N³LO (aN³LO) order was performed by the MSHT group, and publicly released in the MSHT20an3lo PDF set. This accounted for the significant amount of known information about the N³LO results for the PDF evolution, heavy flavour transitions and DIS coefficient functions, while also including approximations for the unknown parts, with corresponding theoretical uncertainties associated with these and included in the PDF fit. In particular, given the amount of known N³LO information available, this allowed for an increased level of accuracy in comparison to previous NNLO PDF determinations. Preliminary work in this direction from the NNPDF collaboration has been presented in [6].

A separate element of the theoretical calculation considered in the PDF analyses of [7–10] relates to the inclusion of electroweak (EW) and in particular QED corrections to the PDF fit. As well as modifying the DGLAP evolution of the partons, these necessitate the inclusion of a photon constituent of the proton, with a corresponding photon PDF. This then enters the calculation of collider processes via photon–initiated channels that will occur. The impact of these, and QED corrections in general, is relatively moderate but cannot be omitted at the percent level of precision required for current LHC physics.

Both of the above elements, namely the inclusion of corrections up to aN³LO in QCD, as well as QED corrections, and the photon PDF, are therefore crucial when providing the highest precision and accuracy PDF fit possible. However, until now these have not been combined in a single fit. In this paper, we rectify this situation, presenting the first combined QED and aN³LO QCD global PDF determination. These are provided in the `MSHT20qed_an3lo` PDF set.

Having accounted for both sets of corrections, we consider the impact on the resulting PDFs as well as the key LHC phenomenological application of Higgs production in gluon fusion. Here, QED corrections are seen to lead to some further mild reduction in the predicted N³LO cross section, on top of the larger reduction we find from aN³LO corrections to the PDFs. We also analyse the impact on $VH$ and Drell Yan cross-sections, finding in this case that QED and aN³LO effects act in opposite directions, with the QED corrections reducing the cross section and aN³LO corrections leading to some increase. Here, an improved perturbative stability (for both QCD and QED PDFs) is seen in comparison to when NNLO PDFs are combined with the N³LO prediction. We in addition address the question of the extent to which QED and aN³LO QCD corrections factorise in terms of their PDF impact. Namely, whether the relative change from including QED corrections is similar at lower orders in QCD to that at aN³LO. Broadly speaking, we find that this is the case.

Finally, we also briefly present in this paper a new leading order (LO) in QCD fit which includes QED corrections. As discussed in e.g. [1] a LO fit is still of use in for example Monte Carlo event generation for LHC physics. In this case, it can be useful to provide a fit that consistently includes a photon PDF, and hence we provide this here, and briefly discuss the PDFs that result from this fit.

The outline of this paper is as follows. In Section 2 we provide a brief overview of the manner in which QED and aN³LO QCD corrections are simultaneously included in the MSHT fit. In Section 3.1 we present the resulting fit quality, and compare to the NNLO case. In Section 3.2 we present the resulting PDFs and the predicted N³LO Higgs production (via gluon fusion), $VH$ and Drell-Yan cross sections. In Section 4 we present the LO QED fit. Finally, in Section 5 we conclude.

## 2  The Combined QED and aN³LO QCD Fit

To produce a QED and aN³LO QCD fit requires a relatively straightforward combination of the theoretical corrections described in [7,8] and [5], respectively. In particular, for the DGLAP evolution of the PDFs we include the splitting functions

$$P_{ij} = \frac{\alpha}{2\pi} P_{ij}^{(0,1)} + \frac{\alpha\alpha_S}{(2\pi)^2} P_{ij}^{(1,1)} + \left(\frac{\alpha}{2\pi}\right)^2 P_{ij}^{(0,2)} \tag{1}$$

$$+ \frac{\alpha_S}{2\pi} P_{ij}^{(1,0)} + \left(\frac{\alpha_S}{2\pi}\right)^2 P_{ij}^{(2,0)} + \left(\frac{\alpha_S}{2\pi}\right)^3 P_{ij}^{(3,0)} \tag{2}$$

$$+ \left(\frac{\alpha_S}{2\pi}\right)^4 P_{ij}^{(4,0)} . \tag{3}$$

Here, the first line corresponds to the known $O(\alpha, \alpha_S\alpha, \alpha^2)$ QED corrections, the second the known up to $O(\alpha_S^3)$ (NNLO) QCD corrections, and the third the aN³LO QCD corrections that are approximately evaluated according to the procedure described in [5]. While the contributions in the first and second lines are included in the MSHT20 NNLO QED fit [8], the second and third are included in the MSHT20 aN³LO fit [5].

Combining QED and aN³LO QCD is then in principle relatively straightforward, and simply requires including all three lines of corrections. In practice, as discussed in [7], the inclusion of QED corrections distinguishes between the up and down type quarks in a manner that purely

QCD DGLAP evolution does not. This therefore requires that the evolution basis of the partons is changed from that used in the default MSHT aN$^3$LO fit (and earlier purely QCD fits) to a set that is separable by charge:

$$q_i^{\pm} = q_i \pm \overline{q_i} \,, \qquad g \,, \qquad \gamma \,, \tag{4}$$

where $i$ denotes any active flavour, $i = u, d, s, c, b$, and the photon $\gamma$ is separated in elastic and inelastic components [7]. The photon PDF is calculated as described in [8], i.e. following a suitable reorganisation the `LUXqed` formalism [11, 12].

As described in [7], this basis requires some modification of the DGLAP splitting kernals used. In particular, the evolution of the $q_i^-$ is not diagonal in flavour space in the manner that the non–singlet quark distributions that define the default MSHT QCD basis are. These evolve according to

$$\frac{\partial q_i^-}{\partial t} = P_{NS}^- \otimes q_i^- + \sum_{j=1}^{n_F} P_{NS}^{\rm s} \otimes q_j^- \,, \tag{5}$$

where $P_{NS}^{\rm s}$ is first non–zero at NNLO in QCD. At N$^3$LO this (as well as $P_{NS}^-$) is also very well determined in [13], and can be safely set to the central value from that analysis. Thus, the evolution of this QED basis proceeds as in the NNLO in QCD case described in [7], but with the QCD splitting functions suitably generalised to aN$^3$LO order as in [5].

The data included in the fit is very similar to that of the public MSHT20aN$^3$LO release [5], but with some additional updates. Namely the ATLAS 8 TeV jets [14] are now included, while the treatment of certain other jet datasets is also altered. In particular, in the original MSHT20aN$^3$LO study [5] the CMS 7 TeV inclusive jet data were taken with $R = 0.5$, rather than $R = 0.7$, which we now take for consistency with other jet data sets, while NLO EW corrections were omitted in the CMS 7 or 8 TeV inclusive jet data, and are now appropriately included. Finally the effect described in [15] (Footnote 7) is also corrected for here. Otherwise, our treatment of EW corrections follows that described in [8].

In terms of the theoretical treatment of the aN$^3$LO ingredients, these remain as in the public MSHT20aN$^3$LO release [5]. We in particular do not include information due to more recent theoretical calculations of the splitting functions and heavy flavour transition matrix elements [16–22] that have become available after the release of this set. This allows us to isolate the impact of including QED corrections with respect to the same theoretical QCD treatment as in the original releases. A full consideration of these updates is beyond the scope of the current study, but upon initial investigation the impact of these newer theoretical ingredients is found in most cases to be small with respect to the aN$^3$LO baseline, with the differences in some limited regions at most of order the PDF uncertainties, which we recall are designed to include a theoretical uncertainty from the unknown ingredients at the time of the release. This issue will be addressed in detail in a future publication.

Finally, we note that the PDF eigenvectors that we provide differ somewhat from those in [5]. We will in particular make the (very good) approximation discussed there that the uncertainties associated with the aN$^3$LO K–factors in hadronic processes are treated as fully decorrelated from the remaining PDF and theory parameters. In [5] PDF eigenvector sets associated with these 10 K–factor eigenvectors were provided, however within the decorrelated approximation the PDFs themselves do not change here. Indeed, upon inspection it is found that the PDF eigenvector sets associated with these K–factors in the MSHT20aN$^3$LO set are extremely close to the central set, and can therefore be dropped from any PDF error analysis, with the central value and uncertainty on the K–factors themselves simply provided in [5]. For convenience, we now drop these entirely, giving 84 eigenvector directions (rather than 104[1]) associated with the

_________________

[1]For the case of the public MSHT20aN$^3$LO (decorrelated K–factor) set therefore using only the first 84 eigen-

QCD partons, and an additional 6 eigenvectors (12 directions) associated with the uncertainty on the photon PDF input, as described in [7,8]. This therefore results in a total of 96 eigenvector directions for the `MSHT20qed_an3lo` PDF set.

# 3  Results

We present results for a range of fits to the datasets described in the previous section. Namely, we consider fits at both NNLO and aN$^3$LO in QCD, and with and without including QED corrections. For those processes where photon–initiated production can be consistently included with the generated photon PDF these are included only in the QED fits. However, other EW corrections to the cross sections are accounted for in the same manner for all fits, see the discussion in [8] for more details.

## 3.1  Fit Quality

We begin by analysing the fit qualities of the various PDF fits. The breakdown of the fit quality for the non–LHC and LHC datasets is given in Tables 1 and 2 respectively, with the total fit quality given at the end of Table 2. In more detail we show: in the second column the fit quality, $\chi^2/N_{\rm pt}$, for the baseline aN$^3$LO + QED fit; in the third column the largest differences in the $\chi^2$ between the QED and QCD only fits at aN$^3$LO order, with a positive value indicating a worse fit quality in the QED case; in the fourth column the same difference as in the third column but at NNLO in QCD; finally, in the fifth column the largest differences between the aN$^3$LO and NNLO fit qualities for both the QCD only and QED fits is shown. In particular for the $\chi^2$ differences we, for clarity, show only those cases for which the difference is greater than one unit, with a positive (negative) difference indicated in red (blue). We note that here an in what follows 'QCD' or 'QCD–only' is used to distinguish the fit from the case where QED effects are included, although as discussed above e.g. appropriate EW corrections are included in all cases.

Starting with the total fit quality, in the aN$^3$LO fit the inclusion of QED effects is seen to give a very small deterioration in the fit quality, by 3.6 for a total of 4534 points. At NNLO a similar deterioration is seen, but by a somewhat larger amount of 17.3. The latter result is qualitatively consistent with the MSHT20 QED study [8], where a slightly larger difference of 24.3 was found at NNLO, which can be explained by the somewhat different dataset and data treatments described in the previous section. Therefore, we can see that the inclusion of aN$^3$LO QCD theory leads to an overall smaller deterioration in the fit quality upon the inclusion of QED corrections, although the QED fit is still very slightly worse overall than the QCD one.

Viewed another way, we find that the improvement in the fit quality in going from NNLO to aN$^3$LO in QCD is by $\sim 209$ in the QCD only fit, but that there is a more significant improvement of $\sim 223$ when QED corrections are included. In other words, the greater improvement in the aN$^3$LO case allows for the deterioration in fit quality that is introduced at NNLO upon the inclusion of QED corrections to be compensated for to a large extent. If we fix the hadronic K–factors to the NNLO values, a very similar level of improvement is seen, with respect to an overall worse fit quality for both the QCD and QED fits. This indicates that the reduction in the level of deterioration is driven by the new information from known N$^3$LO ingredients that enter, rather than the additional $K$–factor freedom in the hadronic cross sections. This is perhaps unsurprising, given that the major impact of QED corrections is on the PDF evolution, for which much is already known at N$^3$LO. We note that for the QCD only fit the improvement at NNLO presented here is $\sim 50$ points greater than that observed in [5]. This is due to the

---

vector directions provides a very good approximation to the full 104 eigenvector case.

| Data set | $\chi^2/N_{pt}$ aN$^3$LO (QED) | $\Delta\chi^2_{aN^3LO}$ QED-QCD | $\Delta\chi^2_{NNLO}$ QED-QCD | $\Delta\chi^2_{QCD,QED}$ aN$^3$LO-NNLO |
|---|---|---|---|---|
| BCDMS $\mu p$ $F_2$ [23] | 182.6/163 | (+6.6) | (+4.0) | (-3.7, -1.1) |
| BCDMS $\mu d$ $F_2$ [23] | 150.7/151 | - | - | - |
| NMC $\mu p$ $F_2$ [24] | 122.6/123 | - | - | (-2.2, -2.4) |
| NMC $\mu d$ $F_2$ [24] | 103.8/123 | - | - | (-10.1, -9.6) |
| NMC $\mu n/\mu p$ [25] | 131.5/148 | (-1.1) | (-1.3) | (+2.5, +2.7) |
| E665 $\mu p$ $F_2$ [26] | 66.6/53 | - | - | ( - , +1.5) |
| E665 $\mu d$ $F_2$ [26] | 63.0/53 | - | - | (+2.9, +3.4) |
| SLAC $ep$ $F_2$ [27, 28] | 31.3/37 | - | - | (-1.3, -1.4) |
| SLAC $ed$ $F_2$ [27, 28] | 22.4/38 | - | - | - |
| Fixed target/HERA $F_L$ [23, 24, 28–31] | 45.3/57 | - | - | (-21.7, -21.6) |
| E866/NuSea $pp$ DY [32] | 218.4/184 | - | - | (-6.5, -6.1) |
| E866/NuSea $pd/pp$ DY [33] | 7.9/15 | - | - | - |
| NuTeV $\nu N$ $F_2$ [34] | 33.3/53 | (-1.5) | - | (-3.3, -4.1) |
| CHORUS $\nu N$ $F_2$ [35] | 28.3/42 | (-1.6) | - | (-1.0, -1.0) |
| NuTeV $\nu N$ $xF_3$ [34] | 33.1/42 | - | - | (+1.1, +1.4) |
| CHORUS $\nu N$ $xF_3$ [35] | 17.7/28 | - | - | - |
| CCFR $\nu N \to \mu\mu X$ [36] | 67.9/86 | - | - | - |
| NuTeV $\nu N \to \mu\mu X$ [36] | 53.7/84 | - | (-1.1) | (-4.3, -4.8) |
| HERA $e^+p$ CC [37] | 51.9/39 | (+1.0) | - | (+1.3, +1.4) |
| HERA $e^-p$ CC [37] | 67.8/42 | (+1.7) | (+1.9) | (-4.8, -4.9) |
| HERA $e^+p$ NC 820 GeV [37] | 84.4/75 | - | - | (-5.4, -5.5) |
| HERA $e^+p$ NC 920 GeV [37] | 472.3/402 | - | (+2.2) | (-35.5, -38.5) |
| HERA $e^-p$ NC 460 GeV [37] | 246.6/209 | - | - | - |
| HERA $e^-p$ NC 575 GeV [37] | 248.6/259 | - | - | (-13.5, -14.3) |
| HERA $e^-p$ NC 920 GeV [37] | 242.6/159 | (+1.0) | (+1.3) | (-1.6, -1.9) |
| HERA $ep$ $F_2^{c,b}$ [38] | 134.8/79 | (+1.5) | (+1.2) | (+5.8, +3.0) |
| DØ II $p\bar{p}$ incl. jets [39] | 116.7/110 | - | - | (-5.5, -7.1) |
| CDF II $p\bar{p}$ incl. jets [40] | 68.8/76 | - | - | (+6.6, +6.5) |
| CDF II $W$ asym. [41] | 18.8/13 | - | - | - |
| DØ II $W \to \nu e$ asym. [42] | 29.9/12 | - | - | (-1.4, -2.4) |
| DØ II $W \to \nu\mu$ asym. [43] | 15.8/10 | - | - | (-1.7, -2.3) |
| DØ II $Z$ rap. [44] | 17.4/28 | - | - | (+1.0, +1.0) |
| CDF II $Z$ rap. [45] | 40.3/28 | - | - | (+3.7, +3.7) |
| DØ $W$ asym. [46] | 11.1/14 | (+1.0) | - | (-1.8, - ) |

**Table 1:** The values of $\chi^2/N_{pt}$ for the non-LHC data sets. The difference in $\chi^2$ between different fits is also shown explicitly, for the cases that the magnitude is larger than 1 point. In particular, the 3rd column corresponds to the difference between the QED and QCD fits at aN$^3$LO, the 4th column corresponds to the difference between the QED and QCD fits at NNLO, and the fifth column corresponds to the difference between the aN$^3$LO and NNLO fits in the QCD, QED cases.

| Data set | $\chi^2/N_{\rm pt}$ aN$^3$LO (QED) | $\Delta\chi^2_{\rm aN^3LO}$ QED-QCD | $\Delta\chi^2_{\rm NNLO}$ QED-QCD | $\Delta\chi^2_{\rm QCD,QED}$ aN$^3$LO-NNLO |
|---|---|---|---|---|
| ATLAS $W^+$, $W^-$, $Z$ [47] | 30.2/30 | - | - | - |
| CMS $W$ asym. $p_T > 35$ GeV [48] | 6.2/11 | (-2.1) | - | (-2.1, -2.1) |
| CMS asym. $p_T > 25, 30$ GeV [49] | 7.4/24 | - | - | - |
| LHCb $Z \to e^+e^-$ [50] | 24.1/9 | - | - | (+1.4, +1.0) |
| LHCb $W$ asym. $p_T > 20$ GeV [51] | 12.4/10 | - | - | - |
| CMS $Z \to e^+e^-$ [52] | 17.6/35 | - | - | - |
| ATLAS High-mass Drell-Yan [53] | 19.4/13 | - | - | - |
| CMS double diff. Drell-Yan [54] | 128.7/132 | - | - | (-16.9, -16.8) |
| Tevatron, ATLAS, CMS $\sigma_{t\bar{t}}$ [55–70] | 13.9/17 | - | - | - |
| LHCb 2015 $W$, $Z$ [71,72] | 103.3/67 | - | (-1.4) | (+1.8, +2.5) |
| LHCb 8 TeV $Z \to ee$ [73] | 28.6/17 | - | - | (+3.3, +3.2) |
| CMS 8 TeV $W$ [74] | 12.5/22 | (-1.1) | - | ( - , -1.6) |
| ATLAS 7 TeV jets [75] | 201.7/140 | (-2.6) | (-4.2) | (-10.8, -9.1) |
| ATLAS 8 TeV jets [14] | 318.6/171 | (-6.2) | (-8.4) | (-11.9, -9.7) |
| CMS 7 TeV $W + c$ [76] | 12.0/10 | - | - | (+4.5, +4.1) |
| ATLAS 7 TeV high precision $W,Z$ [77] | 99.8/61 | (+2.4) | (+2.0) | (-20.4, -20.0) |
| CMS 7 TeV jets [78] | 208.9/158 | - | - | (+5.5, +6.0) |
| CMS 8 TeV jets [79] | 316.8/174 | (+5.1) | (+6.3) | (-7.0, -8.2) |
| CMS 2.76 TeV jet [80] | 109.7/81 | - | - | (+10.3, +9.4) |
| ATLAS 8 TeV $Z$ $p_T$ [81] | 112.1/104 | (+4.0) | (+12.0) | (-87.7, -95.7) |
| ATLAS 8 TeV single diff $t\bar{t}$ [82] | 24.5/25 | - | - | (-1.7, -1.8) |
| ATLAS 8 TeV single diff $t\bar{t}$ dilepton [83] | 1.8/5 | - | - | - |
| CMS 8 TeV double differential $t\bar{t}$ [84] | 23.4/15 | - | - | (+1.3, +1.0) |
| CMS 8 TeV single differential $t\bar{t}$ [85] | 7.6 /9 | - | - | (-1.6, -1.4) |
| ATLAS 8 TeV High-mass Drell-Yan [86] | 65.2/48 | - | - | (+7.7, +7.7) |
| ATLAS 8 TeV W [87] | 57.8/22 | - | - | - |
| ATLAS 8 TeV $W$ + jets [88] | 19.2/30 | - | - | - |
| ATLAS 8 TeV double differential $Z$ [89] | 85.5/59 | (+1.6) | (+1.8) | (+11.2, +10.0) |
| Total | 5323.6/4534 | (+3.6) | (+17.3) | (-209.3, -223.1) |

**Table 2:** The values of $\chi^2/N_{\rm pt}$ for the LHC data sets. The difference in $\chi^2$ between different fits is also shown explicitly, for the cases that the magnitude is larger than 1 point. In particular, the 3rd column corresponds to the difference between the QED and QCD fits at aN$^3$LO, the 4th column corresponds to the difference between the QED and QCD fits at NNLO, and the fifth column corresponds to the difference between the aN$^3$LO and NNLO fits in the QCD, QED cases. The total $\chi^2$ value corresponds to the sum of the individual values shown in Tables 1 and 2.

somewhat different dataset and data treatments described in the previous section, as well as the effect described in [15] (Footnote 7).

Looking in more detail at the changes for the individual datasets, we can see in many cases there are broad similarities between the NNLO and aN$^3$LO results in terms of which data sets see an improvement or deterioration upon the inclusion of QED effects. For example, we see some deterioration in the BCDMS and HERA data, and in the CMS 8 TeV jets, while there is some improvement in the ATLAS 7 and 8 TeV jet data. The difference in the case of the ATLAS and CMS jet data may be connected to the fact that, as observed in [15, 90–92], there is some difference in the pull on the high $x$ gluon between these.

These changes were all qualitatively seen already in [8], with the exception of the ATLAS 8 TeV jet data, which was not included there. As discussed in more detail there, the change in the BCDMS data can be understood from the effect of $q \to q\gamma$ emission which leads to a quicker high-$x$ quark evolution, i.e. mimicking a slightly larger value of $\alpha_S$, which the BCDMS data is known to disfavour. For the other datasets these are sensitive to the high $x$ gluon, which is altered upon refitting by the inclusion of QED effects, due principally to the photon contribution to the momentum sum rule.

The most significant individual difference between the NNLO and aN$^3$LO fits is for the ATLAS 8 TeV $Z$ $p_\perp$ data. Here, we can see that at NNLO a deterioration of $\sim 12$ points is seen upon addition of QED effects. A similar deterioration to this is seen in the previous NNLO analysis [8], and is explainable by the tension that this dataset is known to exhibit with other datasets that are sensitive to the high $x$ gluon. However, at aN$^3$LO it was shown in [5] that this tension was greatly reduced, and the corresponding fit quality to the $Z$ $p_\perp$ data significantly improved. Another effect of this is that, as can be seen from Table 1, while there is still some deterioration in the fit quality to the $Z$ $p_\perp$ upon the inclusion of QED effects, this is now very mild. Indeed, this difference accounts for roughly half of the overall reduction in the deterioration between the NNLO and aN$^3$LO fits. Otherwise, there are some other differences by up to $\sim 2$ points in $\chi^2$, but nothing too significant, and which cumulatively make up the remaining difference.

The above results are also evident in the last column of Tables 1 and 2, where the difference between the aN$^3$LO and NNLO fit qualities, including and excluding QED corrections, is shown. For example, we can see that for the ATLAS $Z$ $p_\perp$ data there is a somewhat larger improvement when QED corrections are included, consistent with the worse fit quality at NNLO. More broadly, there is clearly a similar level of improvement in going to aN$^3$LO with or without QED corrections, with the trends in this largely following that seen in the previous QCD fit [5].

## 3.2 PDFs and Cross Sections

We next consider the impact of including QED corrections on the PDFs. First, in Fig. 1 we show the ratio of the aN$^3$LO PDFs, both including and excluding QED corrections, to the NNLO case without QED corrections. We can see that the broad trends in the pure QCD cases are very similar to those found in [5], which is as expected given the underlying fits are very similar, if not identical. For example, the gluon is enhanced at low $x$ and suppressed in the $x \sim 0.01$ region, while the strangeness is enhanced at high $x$, and the $u_V$ and $d_V$ valence distributions are enhanced at intermediate $x$, see [5] for further discussion. These trends are also very similar once we include QED corrections. In other words, the modifications in the PDFs that come from going from NNLO to aN$^3$LO in QCD are clearly significantly larger than those that come from including QED corrections. Naively, it is sometimes argued that since $O(\alpha(M_Z^2)) \sim O(\alpha_S^2(M_Z^2))$, this would imply that QED corrections are as important as NNLO in QCD. However, this relation only holds at high scales, while for much data in a global fit $Q^2 \sim 10\,\mathrm{GeV}^2$ or less and $\alpha_S$ becomes significantly larger. Also, and more importantly, higher

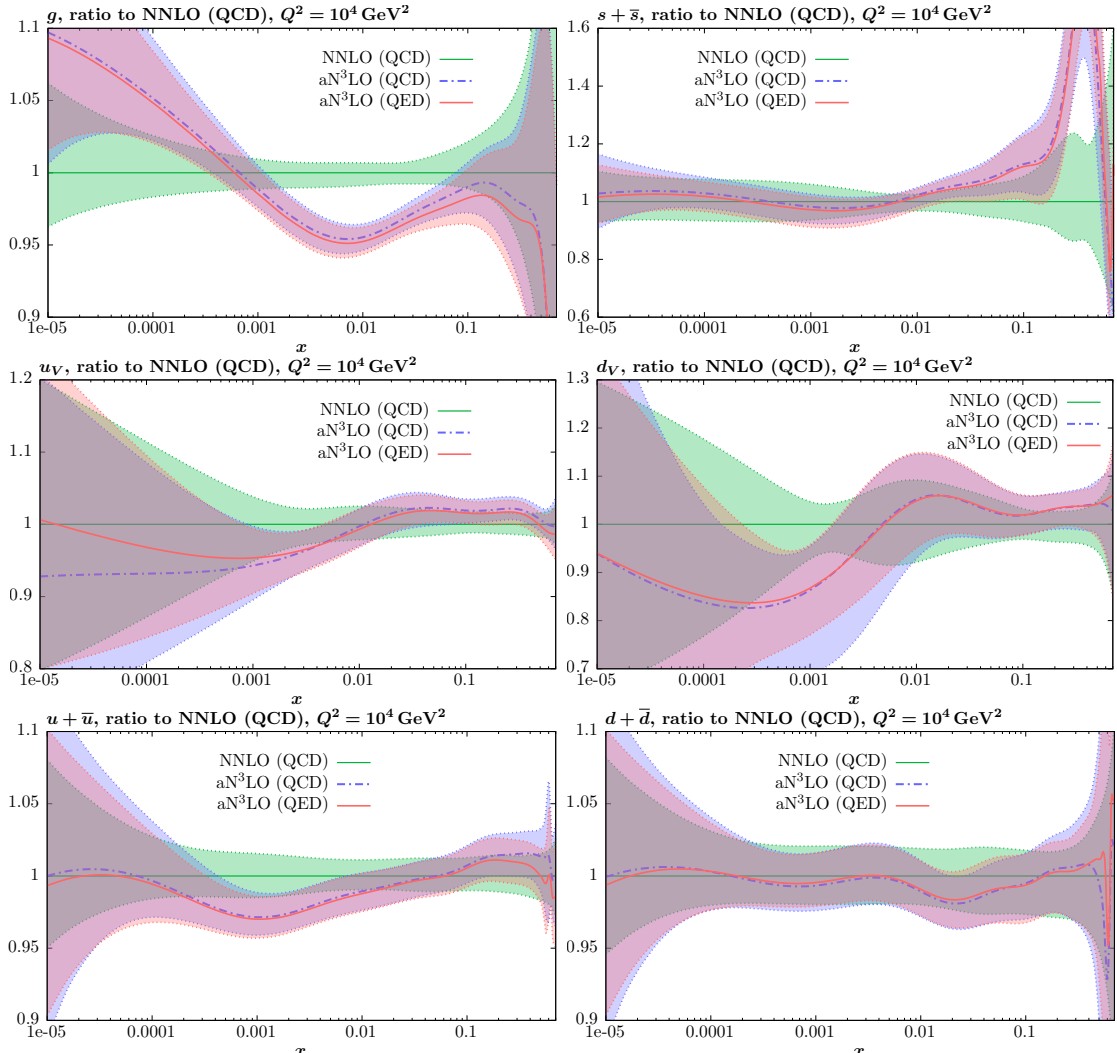

**Figure 1:** PDF ratios of the aN³LO fits, with ('QED') and without ('QCD') including QED corrections to the NNLO fit without QED corrections included.

orders in $\alpha_S$ are accompanied by a variety of higher logarithms in functions of $x$, enhancing the impact of higher orders in QCD. Hence, we see that even aN³LO is still more important than QED corrections in many $x$ regions.

The impact from QED is nonetheless visible on the plots. We can see for example that the gluon is in general slightly suppressed by these corrections, including in the region relevant for Higgs production (a similar effect is seen in other studies [7–10]); we will discuss this further below. Further modifications in the quark sector are also visible, with the impact on the high $x$ up quark singlet, $u + \overline{u}$, being one of the few cases where the impact is in fact similar or larger from including QED corrections than going to aN³LO in QCD.

In Fig. 2 (left) we show the photon PDF in the fits including QED corrections and at aN³LO and NNLO in QCD, and we can see that at aN³LO the photon is $\sim 1 - 3\%$ larger than at NNLO. As the elastic and low scale inelastic input distributions are the same at both orders, this difference can only be driven by the differing QCD partons at the two orders (as well as their QCD evolution), and the impact this has on the perturbatively generated photon PDF, via DGLAP evolution. In the Fig. 2 (right) we therefore show the charge weighted quark/antiquark

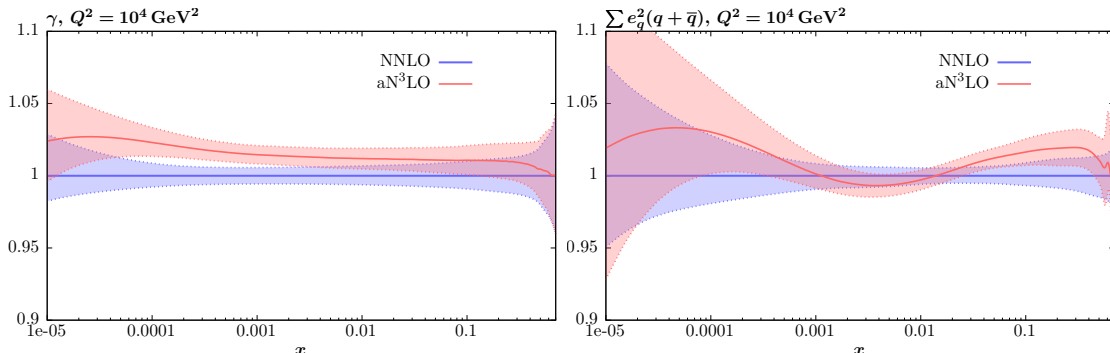

**Figure 2:** PDF ratios of the aN$^3$LO photon and charge weighted singlet to the NNLO fit, with QED corrections included in all cases.

distribution at both orders. The overall difference between the two orders is rather non–trivial, reflecting the changes that occur in the quark sector. At low $x$ the enhancement is driven by the enhancement that is in particular present in the charm and bottom PDFs, as well as the strange to a lesser extent. At intermediate $x$ on the other hand a mild suppression is observed, consistent with the suppression that is in particular seen in the up quark singlet, but also the other quark distributions. At high $x$ the distribution is again enhanced, consistent with the enhancement that is observed across the entire quark sector. The net effect of this, where the charge weighted quark distribution is enhanced over the majority of $x$ by an average of about a couple of percent, is to enhance the corresponding photon PDF by a similar amount.

To investigate the above effects in more details, it is also interesting to see how the relative impact of including QED corrections changes with going from NNLO to aN$^3$LO in the QCD order. This is shown in Fig. 3, and we can see that the broad trends are similar. This is not surprising, as the dominant effects will be very similar irrespective of the QCD order. Namely the reduction in the gluon and strangeness is, as discussed in [7, 8], due to the presence of the photon PDF and the corresponding compensation that is then required in the other partons in order to maintain the momentum sum rule. In addition, the up singlet distribution, $u + \bar{u}$ is reduced at high $x$, due to the impact of $q \to q + \gamma$ emission (for the down case this is largely absent due to the lower electric charge). Both of the above effects will be expected to occur, irrespective of the QCD order, as is observed. Nonetheless, we can see that there are some subtle differences. For example, the reduction in the strangeness, and gluon at low $x$, is somewhat less at aN$^3$LO. There are also some mild differences in the quark sector, in particular the $u_V$, $d_V$ valence distributions.

Another useful way to demonstrate the impact of QED corrections on the aN$^3$LO QCD fit is via their effects on the PDF luminosities at the 14 TeV LHC, as defined in [93], and which are shown in Fig. 4. Here we can see that while again the impact of including QED corrections is in general less than that of going to aN$^3$LO in QCD, the former is nonetheless not negligible. The $gg$ luminosity is broadly suppressed by up to a couple of percent with respect to the QCD only aN$^3$LO fit across the considered mass region, consistent with the impact on the gluon PDF. The $qq$, $q\bar{q}$ and $qg$ luminosities are similarly suppressed, in particular at high mass, again consistent with the change in the quark/antiquark PDFs. The change at the highest mass values is in particular the only region where the impact of QED corrections becomes larger than that of going to aN$^3$LO in QCD. The differences are nonetheless within the luminosity uncertainties. The $\gamma\gamma$ luminosity is also shown, and a consistent level of enhancement is seen as in the photon PDF.

Finally, it is interesting to examine the impact of QED and aN$^3$LO corrected PDFs on a

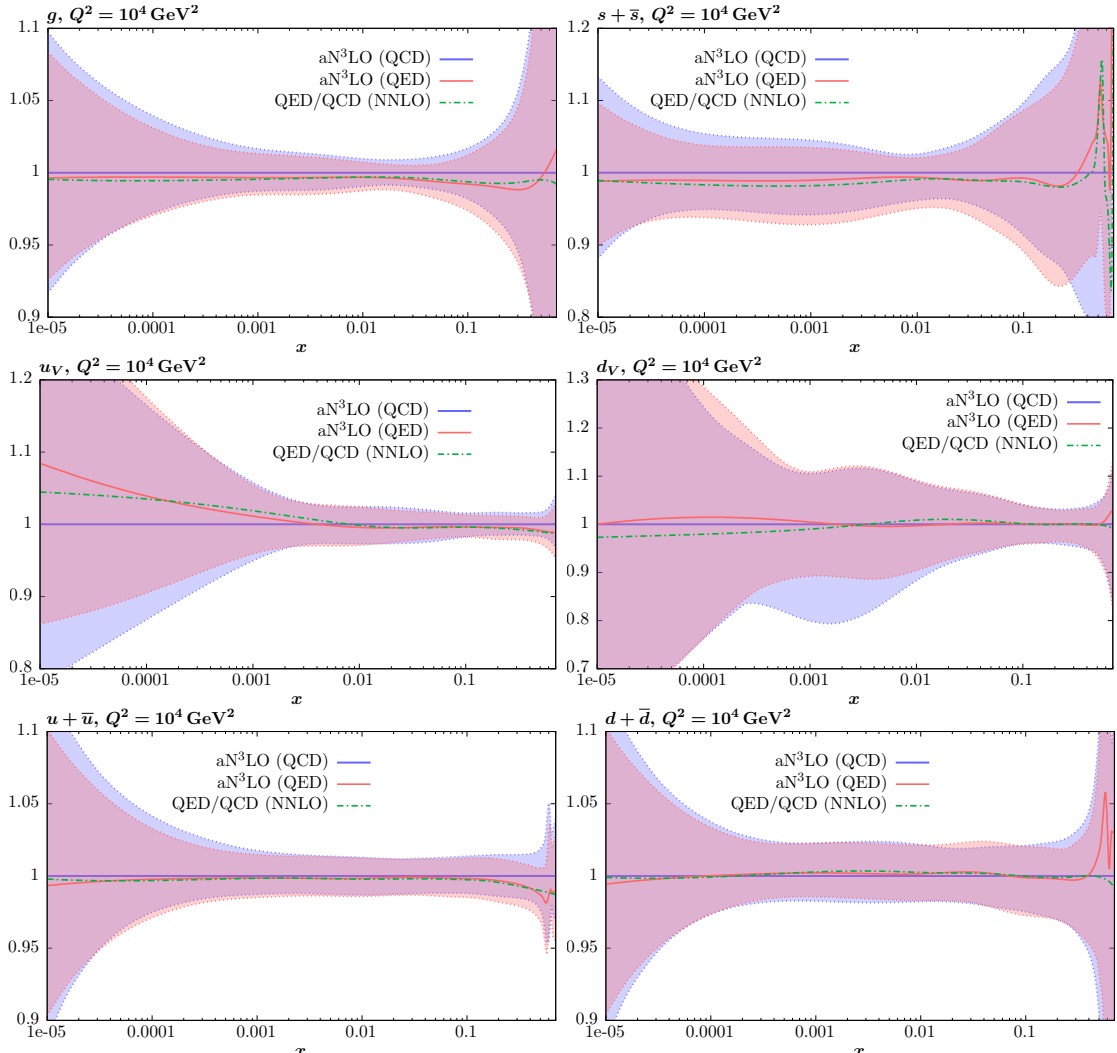

**Figure 3:** PDF ratios of the aN³LO and NNLO fits including QED corrections to that without.

selection of LHC cross sections, where the theoretical calculation is available at N³LO order in QCD. We start with the Higgs production cross section in $gg$ fusion. In this case, we can see in Fig. 4 that the $gg$ luminosity, which is suppressed in the Higgs mass region by the inclusion of aN³LO corrections in the fit, is slightly further suppressed by the inclusion of QED corrections. The production cross sections are plotted in Fig. 5 (top left) and given in Table 3 of Appendix A for a range of different cases, with the corresponding cross sections calculated using `n3loxs` [94]. For the scale choice we take $\mu_F = \mu_R = m_H/2$ and we show results at 14 TeV. We also give the corresponding PDF and 7–point scale variation uncertainties. We note that the purpose here is to to evaluate the impact of PDF effects rather than to compare other theoretical settings. For example, somewhat lower cross section results can be obtained with the `ggHiggs` code [95], due to the inclusion of the bottom and charm Yukawas and also not using the infinite top mass EFT approximation.

We can see that, as seen in [5], the increase that is observed in the cross section in going from NNLO to aN³LO in QCD, when the same (NNLO) PDFs are used, is completely compensated for upon the use of consistent aN³LO PDFs for the latter cross section, with the central value of this now predicted to be somewhat lower than the central value using the NNLO PDFs with the

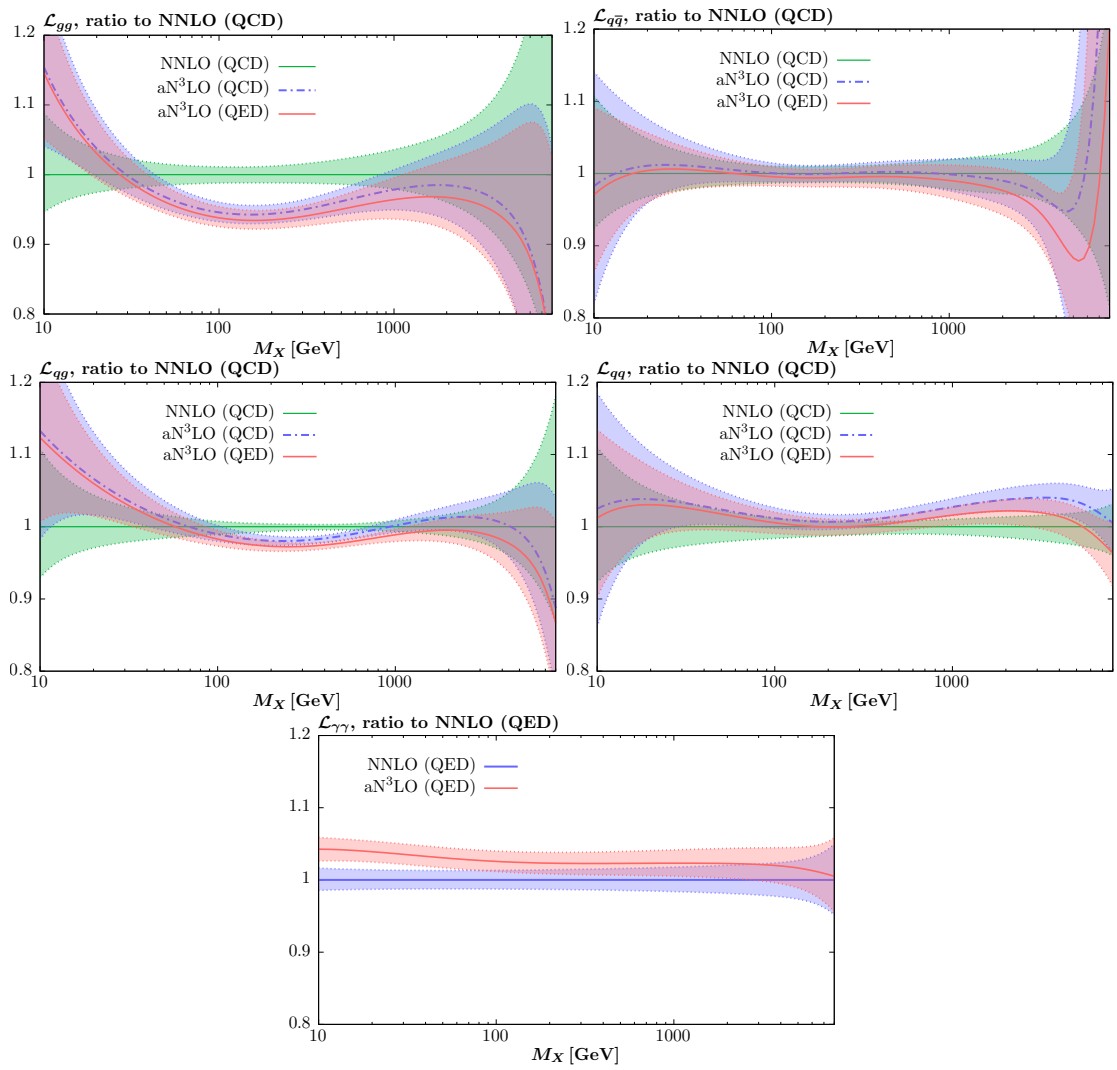

**Figure 4:** Ratio of the PDF luminosities at the 14 TeV LHC for the aN³LO fits, including ('QED') and excluding ('QCD') QED corrections, to the NNLO case with QED corrections excluded.

N3LO cross-section. The inclusion of QED corrections then slightly reduces the cross section further at aN³LO (and NNLO). It should be noted that the above changes are all encompassed in the scale variation uncertainty of the NNLO cross section prediction. The final result, at both aN³LO order in QCD, and including QED corrections in the PDF extraction, then represents the most precise prediction to date with respect to the PDF treatment for the Higgs production cross section via $gg$ fusion.

Next, in Fig. 5 the LHC 14 TeV cross sections for associated $W^{\pm}H$ and $ZH$ production are also shown, again calculated using n3loxs [94]. We can see that the impact of QED is to reduce the cross sections by $\sim 1\%$; a similar reduction was observed in the (related) Drell Yan cross sections in [8] and also below in Fig. 6. This is driven by the reduction that QED effects induce in the $q\bar{q}$ luminosity in the relevant mass region observed in Fig. 4 and driven primarily by the reduction in the strangeness seen in Fig. 3 that occurs due to the inclusion of the photon PDF in the momentum sum rule. The relative reduction is again found to be very similar at NNLO and aN³LO in QCD. The impact of N³LO corrections to the cross section is to reduce the rate by $\sim 1\%$, but this is partly balanced by a small increase in the cross section when aN³LO PDFs are

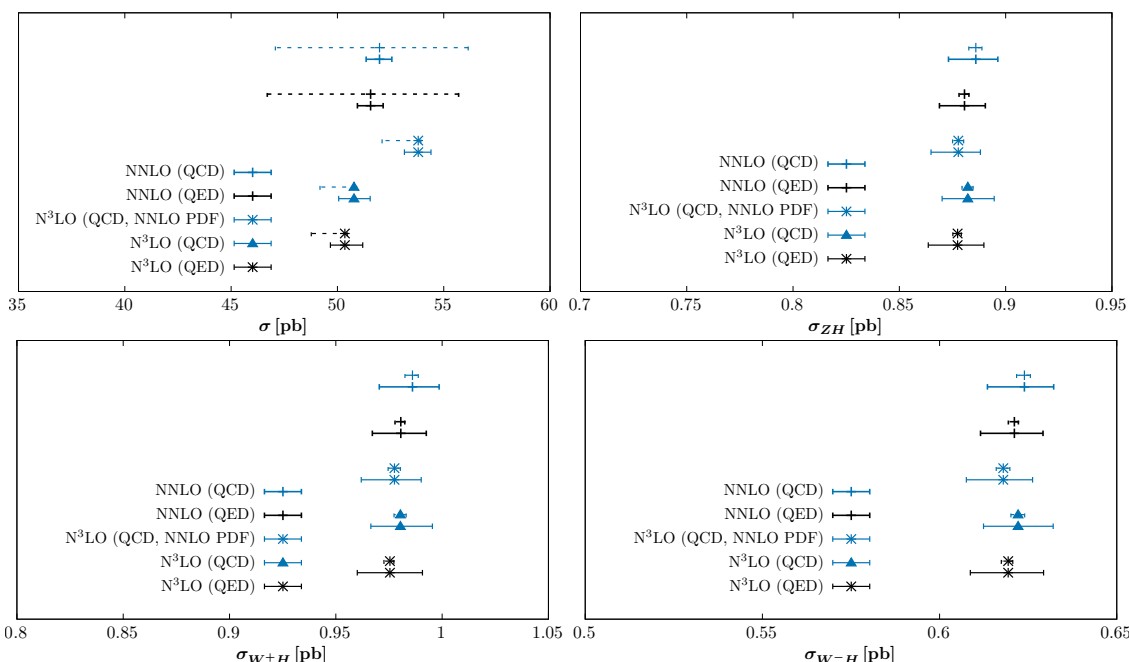

**Figure 5:** Higgs (top left), $ZH$ (top right), $W^+H$ (bottom left) and $W^-H$ (bottom right) cross sections at the $\sqrt{s} = 14$ TeV LHC, calculated with `n3loxs` [94]. The numerical values are given in Tables 3, 4, 5 and 6, respectively (given in Appendix A). The PDF errors are shown by the lower (solid) error bands and the 7–point scale uncertainties by the upper (dashed where large enough to be visible) error bands.

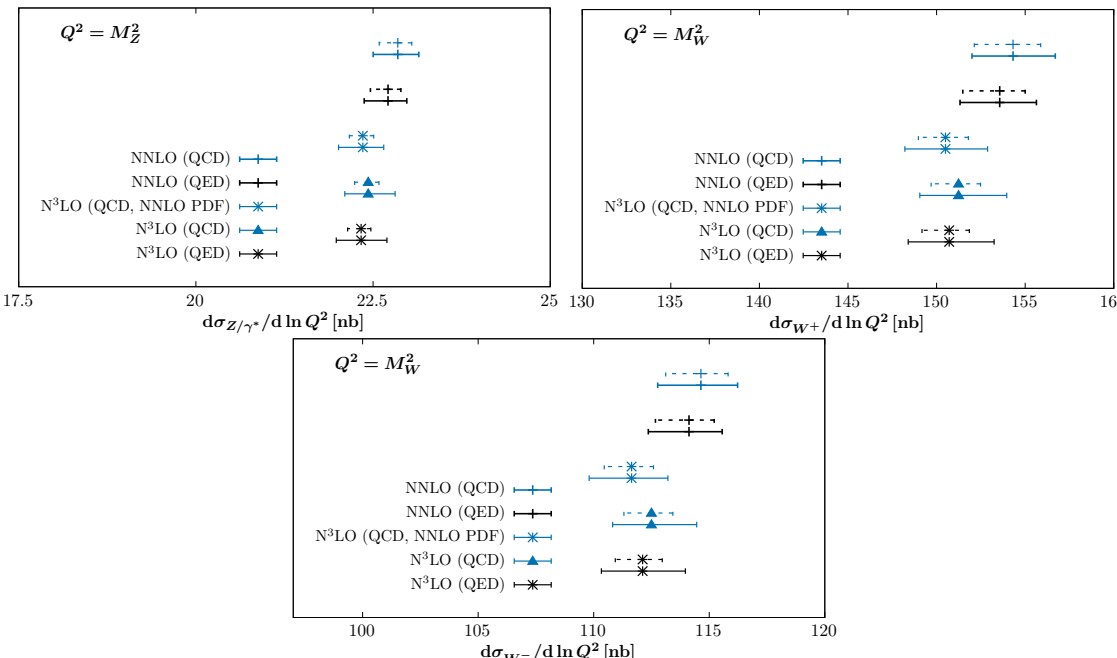

**Figure 6:** The $Z$ (top left), $W^+$ (top right) and $W^-$ (bottom) cross sections at the $\sqrt{s} = 14$ TeV LHC, calculated with `n3loxs` [94]. The numerical values are given in Tables 7, 8 and 9, respectively (given in Appendix A). The PDF errors are shown by the lower (solid) error bands and the 7–point scale uncertainties by the upper (dashed where large enough to be visible) error bands.

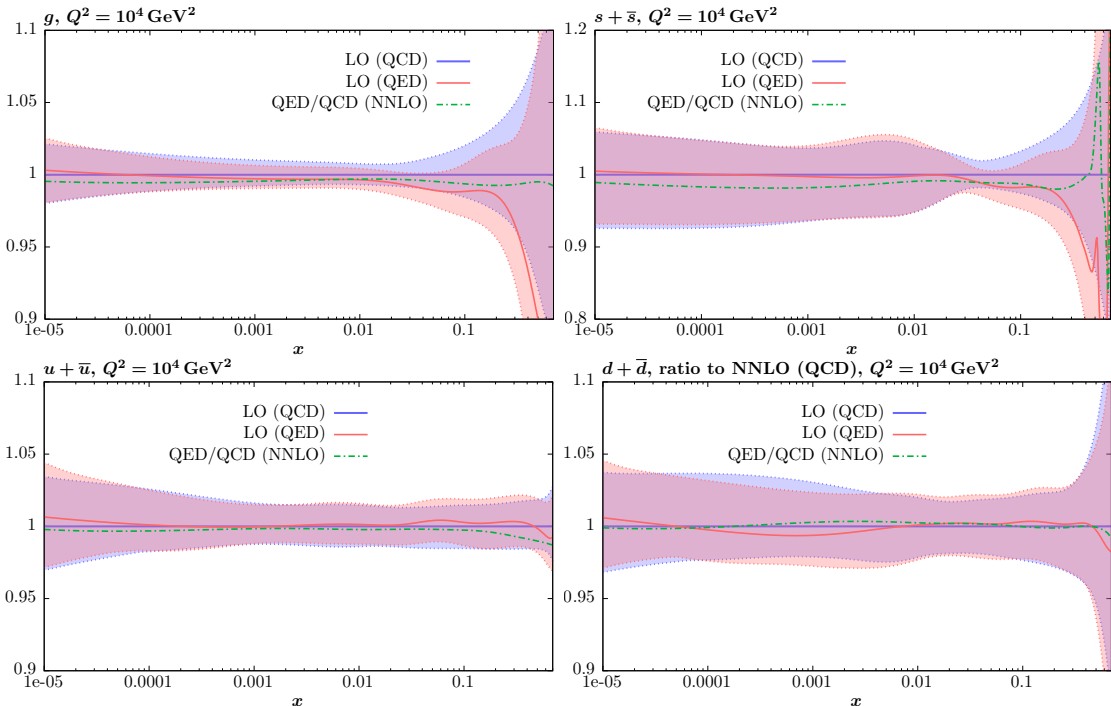

**Figure 7:** PDF ratios of the LO and NNLO fits including QED corrections to that without.

used. This is the net effect of the different changes seen in Fig. 1, that is while the strangeness is increased at aN³LO in the relevant $x$ region, the up and down quark singlet distributions are reduced. As a result of this increase, we find that the perturbative stability is improved, with the NNLO and N³LO (with aN³LO PDFs) results closer to overlapping with the scale variation bands (the rather small size is also observed in [94]). This effect is explicitly verified in the QCD only case, but given the large degree of factorization between QED and QCD corrections observed here, it will also be expected to be present when QED corrected PDFs are used.

In Fig. 6 we show the Drell Yan cross sections, $d\sigma/d\ln Q^2$ at $Q^2 = M_{Z,W}^2$ for $Z/\gamma^*$ and $W^\pm$ production; although this is a somewhat artificial observable, it gives some indication of the relevant trends that we would like to investigate here. Overall, the effect is rather similar to the associated $VH$ case for the relevant boson, as we might expect. That is, we see a reduction in the cross sections upon the inclusion of QED effects in the PDFs, driven by the reduced $q\bar{q}$ luminosity, and a reduction due to the inclusion of N³LO corrections to the cross section, which is in part compensated by the use of aN³LO PDFs. Again, for both QED and QCD PDFs, the aN³LO result leads to improved perturbative stability with respect to the N³LO + NNLO PDF case.

In both the $VH$ and Drell Yan cases, we therefore find that QED and aN³LO corrections compensate each other to some extent, with QED corrections leading to a reduction in the cross section but aN³LO QCD corrections in the PDF leading to an increase. This is in contrast to the Higgs cross section, where both effects lead to a reduction. We note that, as in [8] cross section ratios such as $W^\pm/Z$ are changed less by the addition of both QED and aN³LO effects.

## 4   LO PDF Fit with QED Corrections

In this section, we briefly present the results of a LO fit including QED corrections. To be exact, we also include the same $O(\alpha, \alpha_S\alpha, \alpha^2)$ QED corrections to the DGLAP evolution described in

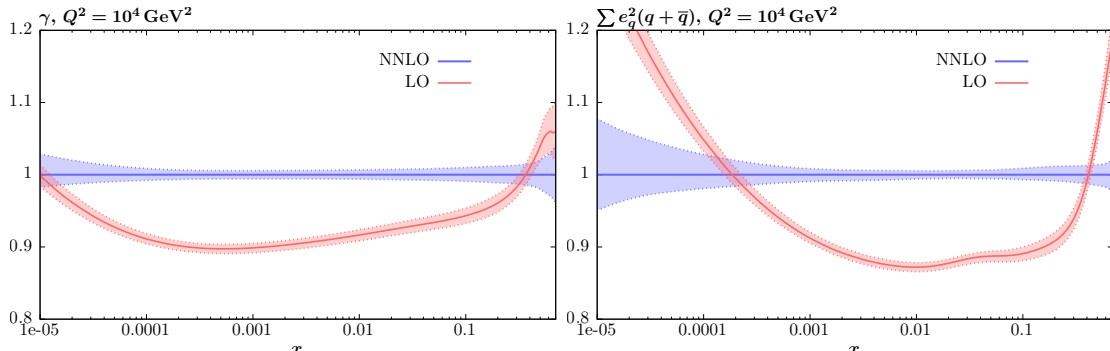

**Figure 8:** PDF ratios of the LO photon and charge weighted singlet to the NNLO fit, with QED corrections included in all cases.

Section 2. While only the $O(\alpha)$ corrections are required in order to consistently include a photon PDF, and the $O(\alpha_S\alpha, \alpha^2)$ corrections are strictly beyond the precision of a LO fit, we continue to include these as for technical reasons this is simpler when performing the fit (and their inclusion is no less accurate than if they were excluded).

As has already been observed in previous MSHT and MMHT fits [1, 96], if a LO fit is attempted with the same parametric freedom as at higher orders, pathological behaviour is generally observed in the extracted distributions. We therefore fix various parameters to avoid this. Namely, the normalisation of the strangeness, $A_{s_+}$, is set to that of the sea, as are 3 of the Chebyshev parameters ($a_{s_+,i}$, with $i = 1, 4, 6$), the high $x$ gluon parameter $\eta_{g_-}$ is fixed, the high $x$ power of the strangeness asymmetry, $a_{s_-}$, is fixed and the sixth Chebyshev of the $\bar{d}/\bar{u}$ is fixed in order to give $\bar{d}/\bar{u} \to 1$ as $x \to 0$. This gives in total 4 eigenvectors fewer than in the default higher fits, that is 28 in total. We in addition exclude the CMS double differential Drell-Yan data [54] from the LO fit, as (see [96]) the lowest mass bin is almost zero at LO due to the specific $p_\perp$ cuts imposed on the leptons.

The fit quality is, as discussed in [1], very poor. For the fit excluding QED corrections we find $\chi^2/N_{\rm pt} \sim 2.59$, very similar to the previous MSHT20 study. When QED corrections are added we find the fit quality deteriorates by $\sim 45$ points, that is with a qualitatively similar trend to the higher order fits, but with a somewhat larger increase.

A brief selection of PDF ratios at LO including QED corrections to that without is shown in Fig. 7, with the corresponding ratio at NNLO also given for comparison. We can broadly see that, as is the case at higher orders, there is a suppression in the gluon and strangeness distributions due to the inclusion of the photon PDF and momentum sum rule constraint. However these reductions are less prominent at intermediate to low $x$ and larger at high $x$. In the up quark singlet only a marginal suppression at higher $x$ is observed.

The photon PDF, and charge weighted quark distributions are shown in Fig. 8. We can see that the LO photon is in general suppressed with respect to the NNLO case, in line with the suppression in the charge weighted quarks. The difference is well outside the quoted uncertainty band, an effect that is observed in earlier LO fits for many of the parton flavours. Given these uncertainties only reflect the underlying experimental uncertainty in the data entering the fit, and how poor the underlying LO fit quality is, this is not entirely surprising. It has long been known, and argued, that PDFs undergo completely qualitative changes when going from LO to NLO due to the first appearance of some divergent terms in $x$ in splitting functions and cross sections, see e.g. [97]. Certainly, in this situation it is far from expected that the uncertainty bands will provide a meaningful estimate.

# 5  Conclusions

In this paper we have presented the first combined QED and aN$^3$LO QCD global PDF determination. We have also presented a new leading order (LO) in QCD fit which includes QED corrections. These are provided in the `LHAPDF6` [98] format at:

https://www.hep.ucl.ac.uk/msht/

as well as on the `LHAPDF` repository, and via the direct links:

MSHT20qed_an3lo
MSHT20qed_lo

As the data and theoretical settings have been updated somewhat since the MSHT20aN$^3$LO analysis [5], we also provide on the website alone the supplementary set corresponding to the QCD–only fit considered here:

MSHT20qed_an3lo_qcdfit

in case the user is interested in isolating the relative impact of QED effects. However, to maintain consistency the public MSHT20aN$^3$LO set remains the official release, with differences between these being in general small and well within PDF uncertainties.

We in addition provide the individual elastic and inelastic photon components, as described in [7, 8], in the sets:

MSHT20qed_an3lo_elastic
MSHT20qed_an3lo_inelastic

MSHT20qed_lo_elastic
MSHT20qed_lo_inelastic

We have considered the impact of combined QED and aN$^3$LO corrections on the resulting PDFs, and found that in general the effect of going to aN$^3$LO in QCD is, as we may expect, rather more significant than that of including QED corrections. Nonetheless, the latter effect remains non–negligible, and must be accounted for given the high precision requirements of LHC physics.

Still, it is interesting to note that in broad terms, what is missed from working only to NNLO in QCD is rather more significant than what is missed by omitting QED corrections to the PDF evolution. In other words, one may call into question the benefit of working with a NNLO QCD + QED fit, if the higher order (approximate) N$^3$LO QCD corrections are omitted. This is of course not always the case, most significantly for those cases where photon–initiated production is important, although as discussed in [8] these are relatively limited for processes of relevance to PDF fits. Moreover, one of course has to bear in mind that strictly speaking aN$^3$LO PDFs are only part of the higher order calculation in any predicted quantity, for which the N$^3$LO cross section is also required.

These possible questions are in any case bypassed by suitably combining aN$^3$LO QCD with QED in the PDF fit, as has been achieved for the first time in this paper. In terms of the PDF impact, we have in addition addressed the question of the extent to which QED and aN$^3$LO QCD corrections factorise. We have shown that indeed they do to good approximation, with

the relative change from including QED corrections being similar at lower orders in QCD to that at aN³LO.

The fit quality has been found to deteriorate by a very small amount at aN³LO upon the inclusion of QED corrections i.e. the $\chi^2$ increases by less than 0.001 per point. This is a rather smaller increase than in the NNLO case, which provides some indication that the higher QCD order provides some further stability in the fit.

The impact on the Higgs cross section in gluon fusion has been examined, and it is found that QED corrections lead to some further mild reduction in the predicted rate at N³LO in QCD. This is however rather less than the reduction found from the inclusion of N³LO QCD corrections in the MSHT PDFs. The relative reduction from QED corrections is found to be similar to that at NNLO, consistent with the factorisation discussed above.

The impact on $VH$ and Drell Yan cross-sections has also been examined. We have found in this case that the QED and aN³LO QCD corrections act in opposite directions, with the QED corrections reducing the cross section and the use of aN³LO PDFs leading to some increase. An improved perturbative stability (for both QCD and QED PDFs) is seen in comparison to when NNLO PDFs are combined with the N³LO prediction.

In summary, we provide a combined aN³LO QCD and QED–corrected PDF set for use by the community, so that they can play a key role in future LHC precision phenomenology. By accounting simultaneously for both QED and aN³LO corrections, an unprecedented level of precision and accuracy in PDF determination has been achieved with respect to the theoretical ingredients entering the PDF fit.

## Acknowledgements

We thank Jamie McGowan, whose invaluable work on the original aN³LO fit provided the groundwork for this study, and to Ilkka Helenius for highlighting the utility of a LO + QED PDF set. TC acknowledges that this project has received funding from the European Research Council (ERC) under the European Union's Horizon 2020 research and innovation programme (Grant agreement No. 101002090 COLORFREE). L. H.-L. and R.S.T. thank STFC for support via grant awards ST/T000856/1 and ST/X000516/1.

## A    Cross Section Results

The cross section and uncertainty values corresponding to Figs. 5 and 6 are given in the tables below.

|  | $\sigma$ [pb] | $\delta$(PDF) | $\delta$(scale) |
|---|---|---|---|
| NNLO (QCD) | 51.98 | $+0.58$ $-0.63$ | $+4.17$ $-4.90$ |
| NNLO (QED) | 51.56 | $+0.59$ $-0.62$ | $+4.14$ $-4.86$ |
| N³LO (QCD, NNLO PDF) | 53.80 | $+0.60$ $-0.65$ | $+0.12$ $-1.70$ |
| N³LO (QCD) | 50.78 | $+0.76$ $-0.72$ | $+0.12$ $-1.60$ |
| N³LO (QED) | 50.35 | $+0.84$ $-0.68$ | $+0.11$ $-1.58$ |

**Table 3:** Higgs cross section via gluon fusion predictions at 14 TeV and their corresponding PDF and scale uncertainties (with the central scale $\mu_F = \mu_R = m_H/2$. Cross sections are calculated with `n3loxs` [94], while the scale uncertainty is calculated using the 7–point variation described in this reference.

|  | $\sigma$ [pb] | $\delta$(PDF) | $\delta$(scale) |
|---|---|---|---|
| NNLO (QCD) | 0.886 | $+0.010$ $-0.013$ | $+0.003$ $-0.003$ |
| NNLO (QED) | 0.881 | $+0.010$ $-0.012$ | $+0.002$ $-0.002$ |
| N³LO (QCD, NNLO PDF) | 0.878 | $+0.010$ $-0.013$ | $+0.003$ $-0.003$ |
| N³LO (QCD) | 0.882 | $+0.012$ $-0.012$ | $+0.002$ $-0.003$ |
| N³LO (QED) | 0.877 | $+0.012$ $-0.014$ | $+0.002$ $-0.002$ |

**Table 4:** $ZH$ cross section predictions at $\sqrt{s} = 14$ TeV and their corresponding PDF and scale uncertainties (with the central scale $\mu_F = \mu_R = M_{ZH}$. Cross sections are calculated with `n3loxs` [94], while the scale uncertainty is calculated using the 7–point variation described in this reference.

|  | $\sigma$ [pb] | $\delta$(PDF) | $\delta$(scale) |
|---|---|---|---|
| NNLO (QCD) | 0.986 | $+0.013$ $-0.016$ | $+0.003$ $-0.004$ |
| NNLO (QED) | 0.981 | $+0.012$ $-0.013$ | $+0.002$ $-0.003$ |
| N³LO (QCD, NNLO PDF) | 0.978 | $+0.013$ $-0.016$ | $+0.003$ $-0.003$ |
| N³LO (QCD) | 0.981 | $+0.015$ $-0.014$ | $+0.003$ $-0.003$ |
| N³LO (QED) | 0.975 | $+0.015$ $-0.015$ | $+0.002$ $-0.003$ |

**Table 5:** $W^+H$ cross section predictions at $\sqrt{s} = 14$ TeV and their corresponding PDF and scale uncertainties (with the central scale $\mu_F = \mu_R = M_{WH}$. Cross sections are calculated with `n3loxs` [94], while the scale uncertainty is calculated using the 7–point variation described in this reference.

|  | $\sigma$ [pb] | $\delta$(PDF) | $\delta$(scale) |
|---|---|---|---|
| NNLO (QCD) | 0.624 | $+0.008$ $-0.010$ | $+0.002$ $-0.002$ |
| NNLO (QED) | 0.621 | $+0.008$ $-0.010$ | $+0.001$ $-0.002$ |
| N³LO (QCD, NNLO PDF) | 0.618 | $+0.008$ $-0.010$ | $+0.002$ $-0.002$ |
| N³LO (QCD) | 0.622 | $+0.010$ $-0.010$ | $+0.002$ $-0.002$ |
| N³LO (QED) | 0.619 | $+0.010$ $-0.010$ | $+0.001$ $-0.002$ |

**Table 6:** $W^-H$ cross section predictions at $\sqrt{s} = 14$ TeV and their corresponding PDF and scale uncertainties (with the central scale $\mu_F = \mu_R = M_{WH}$. Cross sections are calculated with `n3loxs` [94], while the scale uncertainty is calculated using the 7–point variation described in this reference.

|  | $\sigma$ [nb] | $\delta(\text{PDF})$ | $\delta(\text{scale})$ |
|---|---|---|---|
| NNLO (QCD) | 22.85 | $^{+0.30}_{-0.35}$ | $^{+0.20}_{-0.26}$ |
| NNLO (QED) | 22.71 | $^{+0.26}_{-0.34}$ | $^{+0.18}_{-0.25}$ |
| N$^3$LO (QCD, NNLO PDF) | 22.36 | $^{+0.29}_{-0.34}$ | $^{+0.15}_{-0.19}$ |
| N$^3$LO (QCD) | 22.43 | $^{+0.38}_{-0.33}$ | $^{+0.15}_{-0.19}$ |
| N$^3$LO (QED) | 22.33 | $^{+0.29}_{-0.34}$ | $^{+0.15}_{-0.19}$ |

**Table 7:** Cross section prediction for $\mathrm{d}\sigma(\gamma^*/Z)/\mathrm{d}\ln Q^2$ at $Q^2 = M_Z^2$ and $\sqrt{s} = 14$ TeV, with their corresponding PDF and scale uncertainties (with the central scale $\mu_F = \mu_R = Q$. Cross sections are calculated with `n3loxs` [94], while the scale uncertainty is calculated using the 7–point variation described in this reference.

|  | $\sigma$ [nb] | $\delta(\text{PDF})$ | $\delta(\text{scale})$ |
|---|---|---|---|
| NNLO (QCD) | 154.32 | $^{+2.38}_{-2.32}$ | $^{+1.56}_{-2.19}$ |
| NNLO (QED) | 153.56 | $^{+2.07}_{-2.24}$ | $^{+1.44}_{-2.08}$ |
| N$^3$LO (QCD, NNLO PDF) | 150.50 | $^{+2.38}_{-2.29}$ | $^{+1.29}_{-1.53}$ |
| N$^3$LO (QCD) | 151.24 | $^{+2.72}_{-2.18}$ | $^{+1.24}_{-1.55}$ |
| N$^3$LO (QED) | 150.71 | $^{+2.54}_{-2.31}$ | $^{+1.14}_{-1.53}$ |

**Table 8:** Cross section prediction for $\mathrm{d}\sigma(W^+)/\mathrm{d}\ln Q^2$ at $Q^2 = M_W^2$ and $\sqrt{s} = 14$ TeV, with their corresponding PDF and scale uncertainties (with the central scale $\mu_F = \mu_R = Q$. Cross sections are calculated with `n3loxs` [94], while the scale uncertainty is calculated using the 7–point variation described in this reference.

|  | $\sigma$ [nb] | $\delta(\text{PDF})$ | $\delta(\text{scale})$ |
|---|---|---|---|
| NNLO (QCD) | 114.64 | $^{+1.59}_{-1.87}$ | $^{+1.18}_{-1.52}$ |
| NNLO (QED) | 114.12 | $^{+1.44}_{-1.76}$ | $^{+1.09}_{-1.045}$ |
| N$^3$LO (QCD, NNLO PDF) | 111.64 | $^{+1.57}_{-1.84}$ | $^{+0.95}_{-1.18}$ |
| N$^3$LO (QCD) | 112.50 | $^{+1.96}_{-1.68}$ | $^{+0.93}_{-1.19}$ |
| N$^3$LO (QED) | 112.12 | $^{+1.85}_{-1.79}$ | $^{+0.85}_{-1.19}$ |

**Table 9:** Cross section prediction for $\mathrm{d}\sigma(W^-)/\mathrm{d}\ln Q^2$ at $Q^2 = M_W^2$ and $\sqrt{s} = 14$ TeV, with their corresponding PDF and scale uncertainties (with the central scale $\mu_F = \mu_R = Q$. Cross sections are calculated with `n3loxs` [94], while the scale uncertainty is calculated using the 7–point variation described in this reference.

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
