# Peer review of "Combining QED and Approximate N${}^3$LO QCD Corrections in a Global PDF Fit: MSHT20qed_an3lo PDFs"

_SciPost Physics_

## Round 1 · Referee Report · Anonymous (Referee 1) · 2024-3-2

Report

Referee Report

In the manuscript titled: ``Combining QED and Approximate N$^3$LO QCD Corrections in a Global PDF Fit: MSHT20qed_an3lo PDFs''
the authors present the MSHTQED_an3lo PDFs extracted from a fit which combines the known QED corrections including $\alpha$, $\alpha_s \alpha$, and $\alpha^2$ contributions, with QCD corrections including full NNLO plus aN$^3$LO corrections. They do not include recent development in the calculation of the splitting functions at N3LO (and this is explicitly stressed in the text). Nevertheless, using the same information included in their previous MSHTaN$^3$LO analysis, the authors provide an estimate of impact of the QED effects using their original PDF release as a reference. In addition, they present the results of a novel PDF extraction at LO in QCD which includes QED corrections.

The manuscript is timely and well written, and contains new information which will be beneficial for future precision analyses at the LHC. It meets the journal acceptance criteria and I therefore recommend the manuscript for a publication in SciPost. I only have a couple of minor requests which are listed below:

-It would be interesting to compare the MHSTQED_aN3LO PDFs and their errors to those from other groups (e.g., NNPDF and CTEQ) which performed
similar analyses including QED effects but at NNLO in QCD. A figure similar to Fig. 1 in the manuscript could be added illustrating this comparison with a brief explanation in the text. In particular, the authors could show the MHSTQED_aN3LO PDFs vs MHSTQED_NNLO PDFs vs NNPDF4.0QED vs CT18QED. I believe this would increase the value of the manuscript.

-The authors should run a full spell check on the manuscripts as I have spotted a few typos.

After these minor points are addressed, the manuscript is accepted for a publication in SciPost.

Attachment

  • validity: -
  • significance: -
  • originality: -
  • clarity: -
  • formatting: -
  • grammar: -

Author:  Lucian Harland-Lang  on 2024-04-25  [id 4444]

(in reply to Report 1 on 2024-03-02)

We thank the referee for their comments, which we have accounted for in the resubmitted manuscript. A comparison between the different global fit result has in particular been added as Fig. 3.

---

## Round 1 · Referee Report · Anonymous (Referee 2) · 2024-4-12

Strengths

1- The paper is relevant as it presents for the first time the combination of QED corrections and approximate N3LO corrections in the fit of PDFs. 2- The paper is well written and the points made are clear. 3- The paper explores some relevant implications of the results presented.

Weaknesses

1- There are a few additional checks that would strengthen the message of the paper that I have asked the authors to address, if possible.

Report

The Authors present for the first time in the literature the impact of combined QED and approximate N3LO corrections on a fit of PDFs. They find that the effect of the inclusion of approximate N3LO corrections is stronger than the effect of QED corrections and that the two corrections factorise to a good approximations. Both findings are expected, but it is good to see it explicitly.
The paper meets the criteria of relevance, originality and quality required by the Journal and as such I recommend that, after the Authors address the points that I raise in my report, the paper is accepted for publication.

Requested changes

Requested changes: 1-There is typo in the abstract, either “these results” or “this result”. Also, in the abstract the authors mention only the Higgs cross section via gluon fusion, while in the manuscript they also look at V+H and vector boson fusion into Higgs. 2-Given that the aN3LO NNPDF set has been posted to the arXiv, the authors should add a mention to it at the end of the second paragraph of the introduction. Same for the NNPDF4.0 QED set. 3-Towards the end of the introduction, when the authors mention the LO set including QED corrections, or in the section devoted to the set, it would be good to have a few lines on the recommended usage of such set. 4-After Equations (1)-(3) I would expect a short discussion about the possible importance of O(alpha * alphaS^2) corrections that are not included. 5-On page 3, when the authors discuss the way they treat EW corrections, they refer to Ref.[8]. It would be preferable to have a few lines summarising such treatment, so that the paper is more self contained. 6-In Section 2, when the authors discuss the number of eigenvectors, I am curious on whether there is a technical reason why more than 100 eigenvectors are too many? 7- In Fig. 2 the authors show how the photon changes upon increasing the QCD precision of the predictions in the fit from NNLO to N3LO. Given that later in the paper they also show the photon PDF in a LO fit, I think they are missing a chance to show also the photon in a NLO fit and show the perturbative convergence of the photon PDF upon increasing the theoretical accuracy of the fit. Would it be possible to add a plot in which LO, NLO, NNLO and aNNLO photons are displayed simultaneously? 8- On page 10 there is a repeated word “to to”. 9- On the top left plot of figure 5 the x-label should read sigmaH rather than sigma. 10- In 2107.13580 it is discussed the photon PDF’s dependence upon the scale at which the LUX master formula is used to compute the photon PDF. I think that a short discussion of this aspect would be relevant also in this work.

Recommendation

Ask for minor revision

  • validity: high
  • significance: high
  • originality: good
  • clarity: high
  • formatting: good
  • grammar: excellent

Author:  Lucian Harland-Lang  on 2024-04-25  [id 4445]

(in reply to Report 2 on 2024-04-12)

We thank the referee for their useful comments, which we have addressed in the resubmitted manuscript. The changes with respect to these are detailed below:

1) `These results' is correct - PDFs are plural in the previous sentence. The Higgs cross section description has been corrected. 2) These have been added. 3) We have added some additional motivation for why such a LO set can be useful. 4) We must admit that we cannot find any reference to O(alphaalphas^2) corrections in the literature. If these are available, we would be grateful if the referee could point us to them. More generally, even if these were available we can already see that impact of QED effects on the PDF evolution is relatively mild, and hence these O(alphaalphas^2) corrections would be of fairly limited phenomenological impact, justifying their omission. 5) We have added a sentence clarifying this. 6) Having over 100 eigenvectors is not in practice a significant issue, it is just a question of time taken to run when evaluating PDF uncertainties, and given the higher eigenvectors are found to make no difference to the end result, the time spent evaluating these eigenvectors would be redundant. The difference would be relatively small though. 7) While we in principle agree this would be interesting, this would require performing a dedicated NLO fit, and presenting the results of that. Given a NLO QCD + QED PDF set would be of very limited phenomenological relevance, we would prefer not to present this here. However, we have confirmed that the charge weighted quark singlet combination, which is most relevant for determining the difference of the behaviour of the photon PDF with QCD order at high scales, is significantly more convergent at NLO with respect to the NNLO result for the latest MSHT20qed set (i.e. these are consistent within PDF uncertainties), and have added a comment about this to the text, at the end of Section 4. 8) Done, thank you. 9) This is corrected. 10) We have added some discussion of this below (4).

---

## Round 1 · Referee Report · Anonymous (Referee 3) · 2024-4-13

Report

see attached report

Attachment

Recommendation

Ask for major revision

  • validity: -
  • significance: -
  • originality: -
  • clarity: -
  • formatting: -
  • grammar: -

Author:  Lucian Harland-Lang  on 2024-04-25  [id 4446]

(in reply to Report 3 on 2024-04-13)

We thank the referee for their useful comments, which we have addressed in the resubmitted manuscript. The changes with respect to these are detailed below:

1) Corrected. 2) While for clarity we have not shown all of the QCD partons at the input scale, we agree it is informative to show the photon (which is independent of the QCD order in the fit), and have added this to Fig. 2. 3) The dependence of the results on the value of the strong coupling has been discussed in 1907.02750, see Section 4.2.4. The inclusion of QED corrections is found to lead a very small reduction in the best fit value of the strong coupling. The dependence on the strong coupling in the context of the aN3LO corrections has been addressed by us in a very recent publication - 2404.02964. The dependence on the allowed variation of the QED coupling will certainly be marginal within the context of the other uncertainties entering the fit (e.g. on the input photon PDF itself). We have added a paragraph discussing this to Section 2. 4) We have added some clarifying remarks to the first paragraph of Section 3.1 about the Table. 5) Indeed the fit quality is not good according to the textbook definition of things. The reason for this in the case of this specific dataset is not fully understood, though may in part derive from missing resummation effects and/or tensions with other data, but we note that even in the original ATLAS analysis (1612.03016) the fit the ATLAS collaboration themselves perform gives a similar level of fit quality at NNLO. 6) We have added some additional motivation for why such a LO set can be useful in the introduction, where it is more prominent. 7) We have added some further information about the number of eigenvectors to the conclusions to hopefully clarify here. 8) We have clarified this now.

---

## Editorial Decision

resubmitted